# Acute blood loss in mice forces differentiation of both CD45-positive and CD45-negative erythroid cells and leads to a decreased CCL3 chemokine production by bone marrow erythroid cells

Kirill Nazarov[1], Roman Perik-Zavodskii[1‡], Olga Perik-Zavodskaia[1‡], Saleh Alrhmoun[1‡], Marina Volynets[1‡], Julia Shevchenko[1], Sergey Sennikov[1,2]*

1 Laboratory of molecular immunology, Federal State Budgetary Scientific Institution Research Institute of Fundamental and Clinical Immunology, Novosibirsk, Russia, 2 Department of Immunology, Zelman Institute for Medicine and Psychology, Novosibirsk State University, Novosibirsk, Russia

☯ These authors contributed equally to this work.
‡ RPZ, OPZ, SA and MV also contributed equally to this work.
* sennikov@nilkim.ru

## Abstract

Hemorrhage, a condition that accompanies most physical trauma cases, remains an important field of study, a field that has been extensively studied in the immunological context for myeloid and lymphoid cells, but not as much for erythroid cells. In this study, we studied the immunological response of murine erythroid cells to acute blood loss using flow cytometry, NanoString immune transcriptome profiling, and BioPlex cytokine secretome profiling. We observed that acute blood loss forces the differentiation of murine erythroid cells in both bone marrow and spleen and that there was an up-regulation of several immune response genes, in particular pathogen-associated molecular pattern sensing gene *Clec5a* in post-acute blood loss murine bone marrow erythroid cells. We believe that the up-regulation of the *Clec5a* gene in bone marrow erythroid cells could help bone marrow erythroid cells detect and eliminate pathogens with the help of reactive oxygen species and antimicrobial proteins calprotectin and cathelicidin, the genes of which (*S100a8*, *S100a9*, and *Camp*) dominate the expression in bone marrow erythroid cells of mice.

## Introduction

Hematopoiesis in the body ensures the normal functioning of all organs and tissues without exception through the delivery of oxygen by red blood cells, removal of cellular debris and unnecessary substances, elimination of pathogens by immunocompetent cells, tissue remodeling by macrophages, neoangiogenesis, hemostasis, tolerance formation, immune memory, etc. We assume that the regulation of hematopoiesis and the immune response in the hematopoietic organs is carried out by humoral factors produced by erythroid precursor cells—

**Data Availability Statement:** All relevant data are within the manuscript and its Supporting Information files.

**Funding:** This research was funded by Ministry of Higher Education and Science, State Assignment No. 0415-2024-0012 awarded to SS. The specific roles of this author are articulated in the 'author contributions' section. The funders had no role in study design, data collection and analysis, decision to publish, or preparation of the manuscript.

**Competing interests:** The authors have declared that no competing interests exist.

erythroblasts. It has previously been shown that erythroblasts are capable of synthesizing cytokines [1], and the percentage of erythroblasts in hematopoietic organs, in particular in the bone marrow and spleen, is quite significant [2]. Moreover, the levels of cytokine production by erythroblasts are comparable to the levels of cytokine production by peripheral blood mononuclear cells, that is, lymphocytes, and monocytes [3].

In adults, the bone marrow is responsible almost exclusively for physiological erythropoiesis. However, in mice, about 10% of red blood cells are produced in the spleen. Stressful effects (trauma, hemolysis, etc.) activate mechanisms in the body for activating additional so-called foci of extramedullary erythropoiesis in the mouse, in particular in the spleen [2, 4]. In humans, extramedullary erythropoiesis is observed in patients suffering from hematological diseases and tumors. Activation of the glucocorticoid receptor GR is required for stress erythropoiesis and inhibition of differentiation *in vitro*. Interestingly, glucocorticoids activate essentially the same growth inhibition genes in erythroblasts and in lymphocytes, but growth inhibition is counteracted by EpoR/Kit activation, so glucocorticoids promote selective proliferation of erythroid progenitors, supporting erythroblast proliferation and suppressing the proliferation of other myeloid and lymphoid progenitor cells [5]. Fas and FasL are important regulators of immune response and hematopoiesis. Although Fas is expressed on human erythroblasts at all stages of terminal differentiation, only immature erythroblasts die apoptosis by Fas-way. FasL is not expressed until late stage differentiation and orthochromatic erythroblasts exhibit Fas-based cytotoxicity toward immature erythroblasts. However, erythropoietin receptor signaling downregulates erythroblast Fas and FasL during stress thereby preserving the erythropoietic response to stress [6, 7]. Interestingly, Fas overexpression has been detected in CD34+ stem cells from aplastic anemia patients, which may contribute to the suppression of hematopoiesis in this severe disease [8]. Also, it has been shown that erythroferrone protein is the main erythroid regulator of hepcidin, the homeostatic hormone controlling plasma iron levels and total body iron. After stimulation, differentiating erythroblasts in the bone marrow and spleen rapidly increase ERFE production in an EPO-Stat5 dependent manner. ERFE is secreted into the circulation and acts directly on the liver to repress hepcidin. ERFE-mediated hepcidin suppression in turn increases iron availability for new red blood cells synthesis [9, 10]. Acute blood loss (ABL), accompanied by most injuries, represents a request to the hematopoietic organs for new blood cells of all cell types. Post-acute blood loss myeloid and lymphoid cells have been studied in detail, but the immunological changes in the erythroid cells during blood loss have been studied less widely, which, in fact, is what our study was aimed at—to expand the understanding of the role of erythroid cells and to decipher their immune mechanisms of action in the post-acute blood loss period.

## Materials and methods

### Mice

We obtained male mice ♂F1 CBA×C57Bl6 3–5 months old from the vivarium of the Institute of Cytology and Genetics and used them in all experiments. Mice lived in conventional vivarium conditions with water and food access ad libitum, under the natural dark/light cycle. All experiments using mice were approved by the local ethics committee of RIFCI.

### Murine acute blood loss model

Acute blood loss was performed as follows: In mice under isoflurane anesthesia (Aerran, Baxter), ~0.5–0.8 ml of blood was collected from the retro-orbital sinus using a glass capillary with a pointed tip. The onset of surgical anesthesia and recovery from anesthesia for each animal

were monitored. Bone marrow and spleens were collected on the third day after the start of the experiment. Similar intact animals were used as control (conditional norm).

## Mononuclear cell isolation

We harvested femurs (normal bone marrow – $n = 4$, post-ABL bone marrow – $n = 4$) and spleens (normal spleen – $n = 4$, post-ABL spleen – $n = 4$) from mice aseptically. We obtained bone marrow cells by washing the bone marrow canal using PBS. We obtained splenocytes by homogenizing the whole spleen in a glass homogenizer. We centrifuged splenocytes in density gradient Ficoll-Urografin (ro = 1,119 g\cm^3) for 30 min at 322 RCF and washed them twice in PBS to delete RBC and granulocytes.

## Flow cytometry

We washed $0{,}5*10^6$ cells in PBS with the addition of 0,09% $NaN_3$ and stained them with the antibodies according to the manufacturer's protocols. We used Pacific Blue™ anti-mouse TER-119/erythroid Cells Ab #116207, PE anti-mouse/human CD44 Ab #103024, FITC anti-mouse CD45 Ab #304006, and APC anti-mouse CD71 Ab #113820 (Biolegend, San Diego, California, United States). We then washed the cells after 30 minutes of incubation in the dark with 0,5 ml PBS with the addition of 0,09% $NaN_3$. We added 7AAD to all samples right before cytometry. We conducted flow cytometry on the Attune NxT flow cytometer (Thermo Fisher Scientific, Waltham, Massachusetts, United States). We manually gated Ter-119[+] cells as follows: we gated cells from all events, gated singlets from the cells, gated alive cells from the singlets, and gated Ter-119[+] erythroid cells (Scheme 1) in conventional gating software (for Attune NxT) and exported them as.fcs files.

## Flow cytometry data analysis

Ter-119[+] erythroid cell.fcs files were then transformed into.csv files using *fcsparser* library for Python 3. Csv files containing flow cytometry data were *arcsinh*-transformed, *fdaNorm*-normalized and exported as.fcs files with the R script published by Melsen et al [11]. We identified erythroid cell clusters i.e. stages of erythroid differentiation using the prior knowledge that CD71 expression is the highest at the proerythroblast stage and then continuously decreases until it disappears at the reticulocyte stage, Ter-119 is present on all mouse Ter-119[+] erythroid cells. We then exported cluster cell count data into GraphPad Prism 10.2.3. We used either one-way ANOVA with Tukey correction for multiple testing (for the analysis of Ter-119[+] erythroid cells in bulk) or two-way ANOVA with Tukey correction for multiple testing for our statistical analyses (for percentage analysis of differentiation stages of Ter-119[+] erythroid cells).

## Magnetic separation

We performed magnetic separation of mononuclear splenic cells and marrow cells using anti-Ter-119-biotinylated antibodies (#113803, Biolegend, San Diego, California, United States) and streptavidin-linked magnetic beads (#480015, Biolegend, Biolegend, San Diego, California, United States) according to the manufacturer's protocols (MojoSort™ Streptavidin Nanobeads Column Protocol – Positive Selection, accessed on 13.04.2024) to obtain Ter-119[+] erythroid cells.

## Cell viability staining

We measured Ter-119[+] erythroid Cells' viability on a Countess 3 Automated Cell Counter (Thermo Fisher Scientific, Waltham, Massachusetts, United States) according to the

manufacturer's protocols using Trypan Blue. Trypan Blue staining showed > 95% viability for the magnetically separated Ter-119$^+$ erythroid cells.

## Total RNA extraction

We isolated total RNA from 500,000 Ter-119$^+$ erythroid cells with the Total RNA Purification Plus Kit (Norgen Biotek, Thorold, ON, Canada). We measured the concentration and quality of the total RNA in each sample on a Qubit 4 (Thermo Fisher Scientific, USA). We froze the total RNA at −80°C until the gene expression analysis.

## Gene expression profiling

We performed gene expression profiling using the NanoString nCounter SPRINT Profiler analytical system using 100 ng of total RNA from each Ter-119$^+$ erythroid cell sample. We used the nCounter Mouse Immunology v1 panel (561 immunity-related genes, 15 housekeeping genes, 6 positive and 8 negative controls) to analyze the total RNA samples. The samples (normal bone marrow – $n = 3$, post-ABL bone marrow – $n = 3$) were subjected to a 20 h hybridization reaction at 65°C, where 5–14 μL of total RNA was combined with 3 μL of nCounter Reporter probes, 0–9 μL of DEPC-treated water, 12 μL of hybridization buffer and with 5 μL of nCounter capture probes (total reaction volume = 36 μL).

After the hybridization of the probes to targets of interest in the samples, the counts of target molecules were determined on the nCounter digital analyzer. We performed normalization and QC in nSolver 4 using added synthetic positive controls and the *Gapdh*, *Rpl19*, *Ppia*, *Oaz1*, *Eef1g*, *Polr2a*, *G6pdx*, *Gusb*, *Sdha*, *and Alas1* housekeeping genes included in the panel. We then performed background thresholding on the normalized data to remove non-expressing genes. We determined the background level as the mean of the POS_E controls and removed the genes that were below the background level in all samples.

We log$_2$-transformed the gene expression counts data. We then performed differential gene expression analysis using multiple t-tests (we considered Fold Change > 2 or Fold Change < −2 and $q$-values < 0.01 statistically significant) and created a Volcano plot in GraphPad Prism 10.2.3. We created a heat map of the up-regulated and key erythroid cell genes via Bioinfokit [12] and performed Gene Ontology Biological Process overrepresentation analysis of the up-regulated differentially expressed genes via GSEApy [13].

## Cell culturing

We cultured the magnetically sorted cells in the X-VIVO 10 serum-free medium with the addition of x1 Insulin-Transferrin for 24 h at a concentration of 1 million per mL of the medium in order to support their viability and measure culture media cytokines afterward.

## Conditioned media harvesting

We separated Ter-119$^+$ erythroid cell conditioned media from cells after 24 h of culturing. We did the separation by centrifugation at 1500 rpm for 10 minutes, transferred the Ter-119$^+$ erythroid cell conditioned media into new 1,5 mL tubes with the addition of BSA up to the total concentration of 0,5%, and froze the cells' conditioned media at −80°C until the cytokine quantification.

## Cytokine quantification in culture medium using BioPlex

We prepared 50 μL of each Ter-119$^+$ erythroid cell conditioned media sample ($n = 4$ for each bio-group) for a cytokine quantification with a Bio-Plex Pro mouse cytokine 23-Plex assay

(#M60009RDPD, BioRad, Hercules, California, United States) according to the manufacturer's recommendations and analyzed them on the Bio-Plex 200 instrument.

We log$_2$-transformed the Bio-Plex cytokine data using Pandas and then performed multiple T-tests with FDR correction for differential cytokine production analyses (we considered Fold Change > 4 or Fold Change < −4 and $q$-values < 0.05 statistically significant) and created the Volcano plots in GraphPad Prism 10.2.3. We created the heatmap of the profiled cytokine concentrations via Bioinfokit [12].

## Results

### Acute blood loss leads to a decrease in the absolute number of Ter-119$^+$ erythroid cells in the hematopoietic organs of mice

In this work, we analyzed the content of mouse Ter-119$^+$ erythroid cells in the bone marrow and spleen under normal conditions and after acute blood loss. The analysis showed that the content of Ter-119$^+$ erythroid cells in the mouse bone marrow after blood loss is significantly reduced in comparison with intact animals. Also, during blood loss, the content of Ter-119$^+$ erythroid cells in the spleen is significantly lower in comparison with intact animals (Fig 1).

### Acute blood loss leads to stimulation of terminal differentiation of both CD45-negative and CD45-positive Ter-119$^+$ erythroid cells

Then, by applying HSNE (Hierarchic Stochastic Neighbor-Embedded) dimensionality reduction to Arcsinh-transformed and fdaNorm-normalized flow cytometry parameters (FSC, SSC, CD44, CD45, CD71 and Ter-119) of the Ter-119$^+$ erythroid cells, we found clusters corresponding to successive stages of differentiation of Ter-119$^+$ erythroid cells in both branches of murine erythropoiesis–CD45$^-$ and CD45$^+$ (Fig 2).

Stage-by-stage analysis of Ter-119$^+$ erythroid cells showed the presence of statistically significant changes in the structure of erythron. We found that ABL leads to: 1) a significant increase in the relative number of CD45$^+$ proerythroblasts, CD45$^+$ basophilic erythroblasts, CD45$^+$ polychromatic erythroblasts and CD45$^-$ basophilic erythroblasts in the bone marrow, 2) a significant decrease in the relative number of CD45$^+$ and CD45$^-$ reticulocytes, CD45$^-$ polychromatic and CD45$^-$ orthochromatic erythroblasts in the bone marrow, 3) a significant increase in the relative number of CD45$^+$ proerythroblasts, CD45$^+$ basophilic erythroblasts, CD45$^+$ polychromatic erythroblasts and CD45$^-$ basophilic erythroblasts in the spleen, 4) a significant decrease in the relative number of CD45$^+$ orthochromatic erythroblasts, CD45$^+$ reticulocytes, CD45$^-$ polychromatic and orthochromatic erythroblasts and CD45$^-$ reticulocytes in the spleen (Fig 3).

### Acute blood loss leads to the down-regulation of the MHC Class II antigen presentation genes in Ter-119$^+$ erythroid cells

Then we analyzed the expressed genes by the Ter-119$^+$ erythroid cells from the bone marrow cells after the acute blood loss and in normal conditions using the NanoString method on the Sprint platform using the Mouse Immunology V1 panel. The studied Ter-119$^+$ erythroid cells expressed the signature genes of erythroid cells *Tfrc*, *Tal1*, and *Cd36* in a large number of copies, which shows the high purity of the magnetically-sorted cells. Among the genes expressed by Ter-119$^+$ erythroid cells, expression of the cytokine and chemokine genes (in order of decreasing copy number): *Ccl3*, *Tgfb1*, *Il1b*, *Cxcl12*, *Ccl2*, *Mif*, and *Il16*, as well as the antimicrobial response genes *S100a9*, *S100a8*, and *Camp* was detected (Fig 4A and 4B).

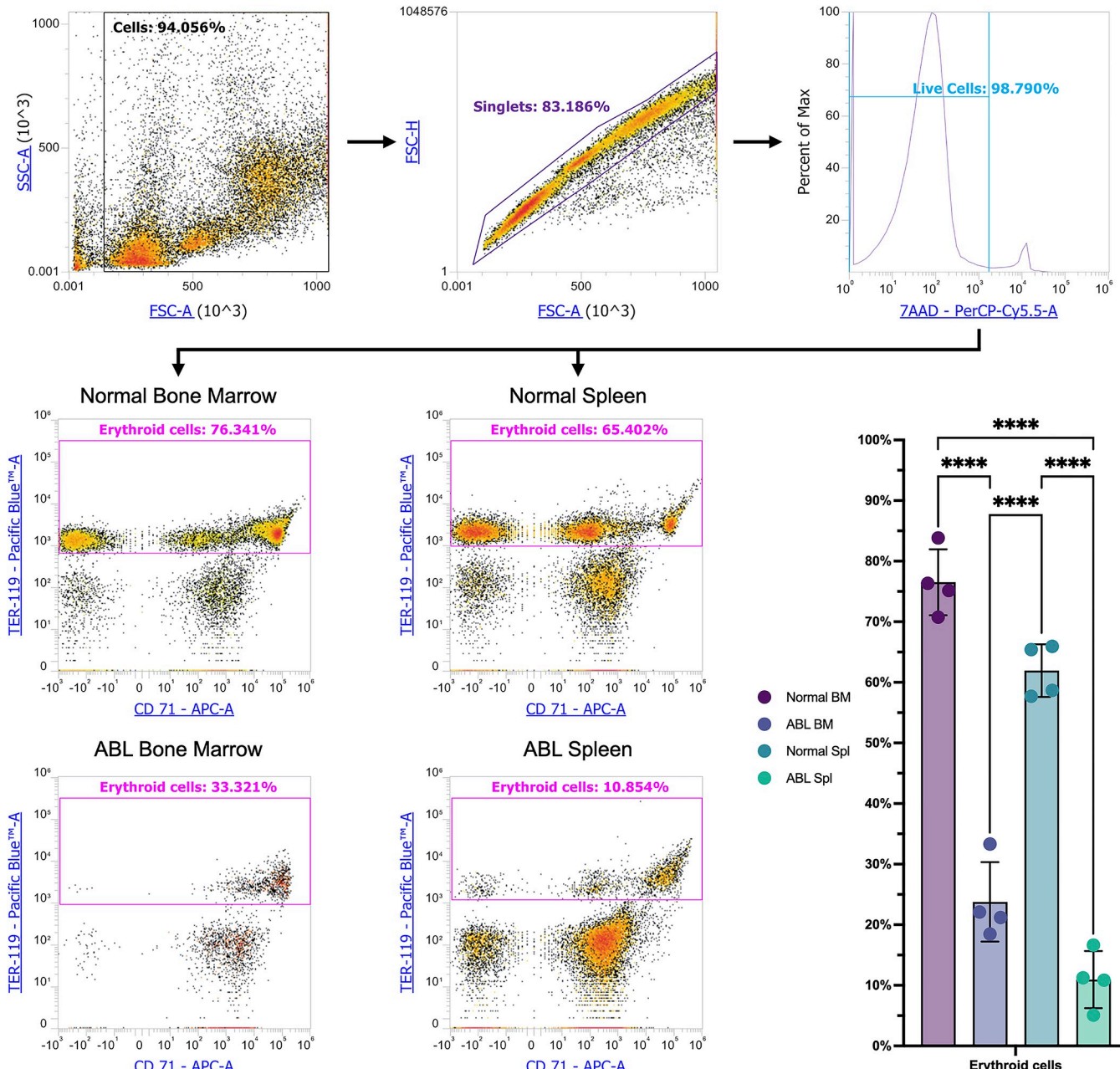

**Fig 1. Percentages of Ter-119$^+$ erythroid cells in the bone marrow and in the spleen in normal condition and after ABL, \*\*\*\* depict statistically significant differences (*q*-values < 0.00005).**

We found statistically and biologically significant changes in gene expression (Fig 4A). We found that after the acute blood loss, Ter-119$^+$ erythroid cells from the bone marrow had higher expression of the genes: *Csf3r*, *Cdkn1a*, *Ctsg*, *Hlx*, *Clec5a*, and *Ltb4r1*; and significantly lower expression of the genes: *Cxcl12*, *Il1r2*, *Ccr2*, *Psmc2*, *Tnfaip3*, *Cd2*, *Cd79b*, *Cd55*, *Cd24a*, *C1qbp*, *Pml*, *Ms4a1*, *Itga4*, *Itgb1*, *Ptpn2*, *Ctnnb1*, *Irgm1*, *H2-Aa*, and *H2-Ab1* (Fig 4A, Table 1).

We also decided to analyze the biological processes occurring in Ter-119$^+$ erythroid cells after the acute blood loss by analyzing invariant (non-up- or down-regulated genes) and up-regulated genes using Gene Ontology Biological Process (GO BP) Overrepresentation Analysis

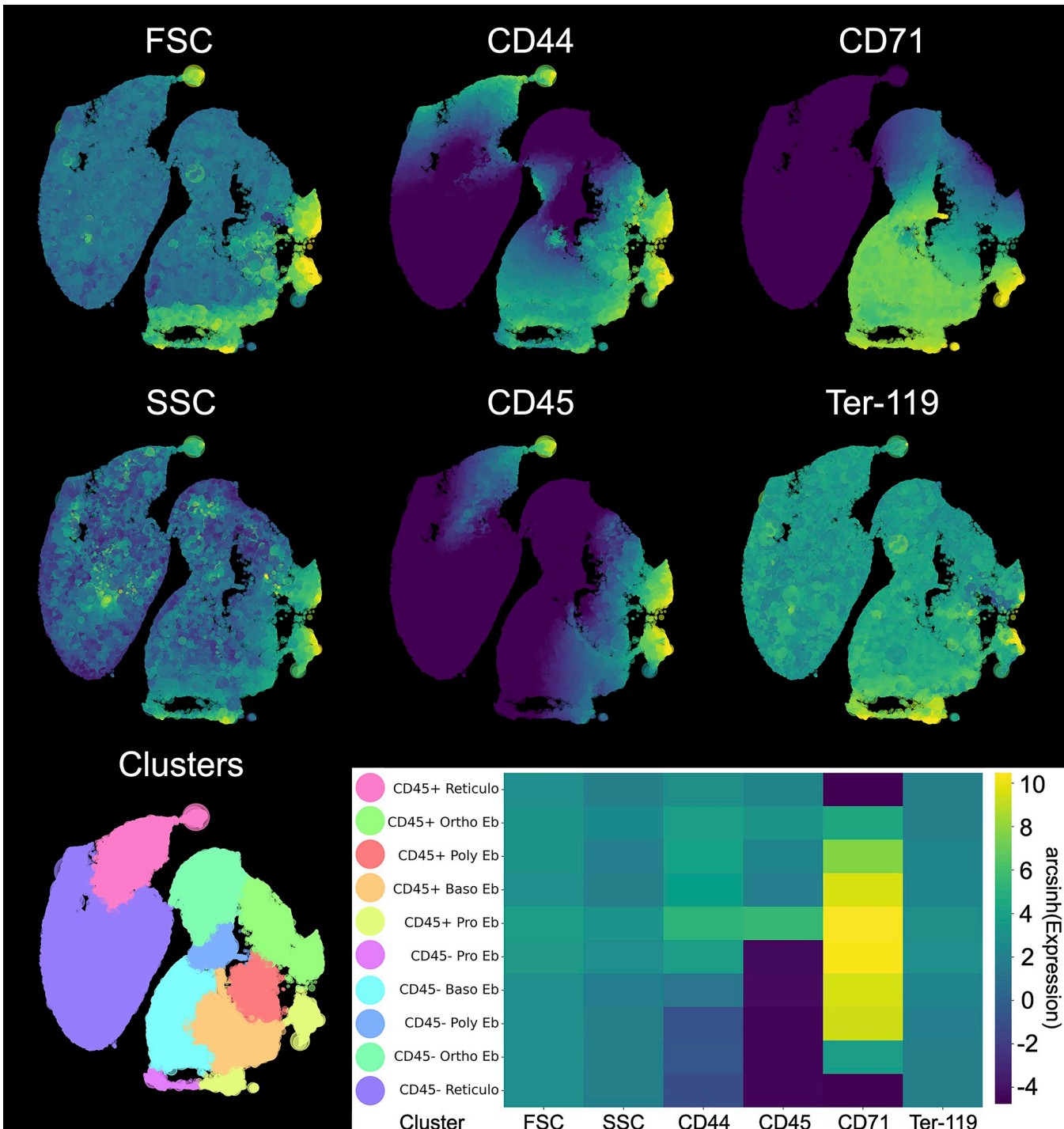

**Fig 2. Integrated HSNE plots of erythroid cells from the bone marrow and the spleen.** Integrated HSNE plots of erythroid cells from the bone marrow and the spleen overlaid with marker expression (see Fig 1 for gating)—the purple color represents the absence of the marker expression whereas the yellow color represents the maximum of the marker expression. Clusters are color-labeled in accordance with the heatmap. The heatmap shows mean hyperbolic arcsine-transformed marker expression values. The yellow color represents the highest standardized expression, and the purple color represents the lowest standardized expression. Pro stands for proerythroblasts, Baso stands for basophilic erythroblasts, Poly stands for polychromatophilic erythroblasts, Ortho stands for orthochromatophilic erythroblasts, and Retic stands for reticulocytes.

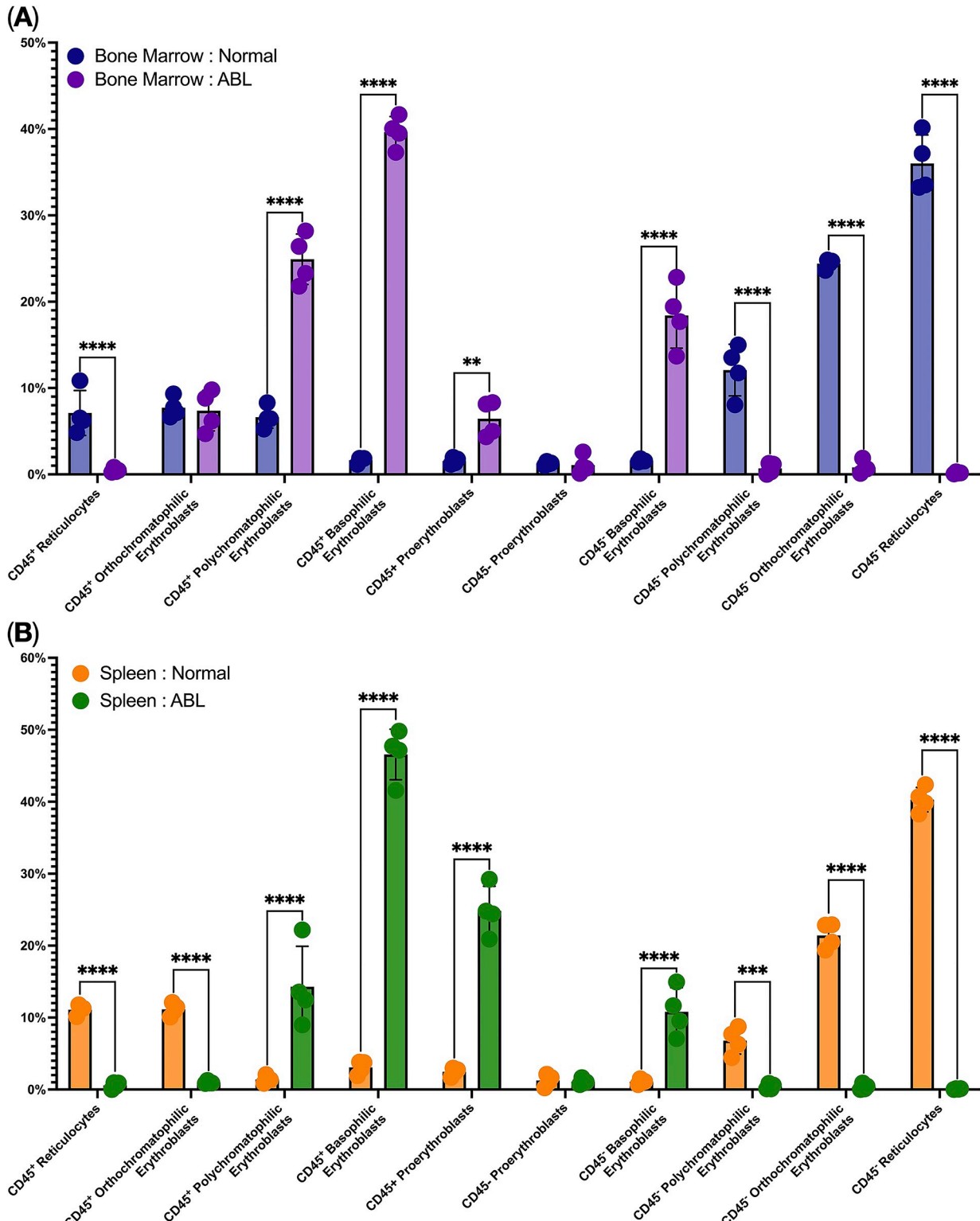

**Fig 3. Percentages of Ter-119$^+$ erythroid cells at the sequential stages of differentiation in the bone marrow and in the spleen in normal condition and after the ABL, **, ***, **** depict statistically significant differences (*q*-values < 0.005, *q*-values < 0.0005, and *q*-values < 0.00005 respectively).**

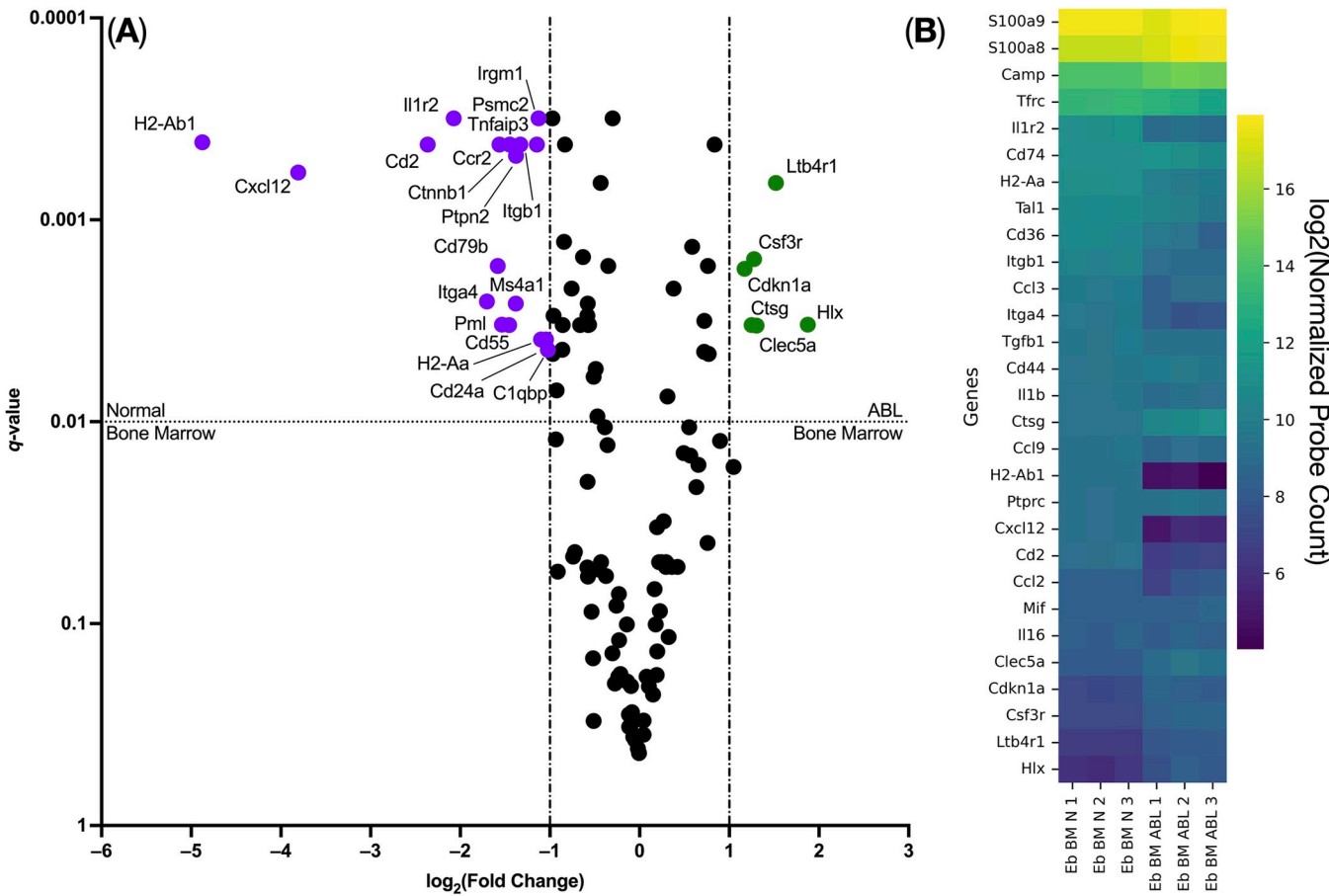

**Fig 4. Differential gene expression analysis of the post-ABL erythroid cells.** (A) Volcano plot of post-ABL Ter-119+ erythroid cells from the bone marrow (Eb BM ABL) versus normal Ter-119+ erythroid cells from the bone marrow (Eb BM N); (B) heat map of the up-regulated and key erythroid cell genes.

and found enrichment in GO BP terms associated with reactive oxygen species synthesis and myeloid cell activation (Fig 5, Table 2).

Analysis of the down-regulated genes revealed that normal bone marrow murine erythroid cells had significantly higher gene expression of the genes that belong to the MHC Class II antigen presentation GO BP Biological Process compared with post-ABL erythroid cells.

## Ter-119+ erythroid cells secrete the chemokines CCL2, CCL3 and the cytokine IL1b

We also studied the cytokine secretion of Ter-119+ erythroid cells in the bone marrow and spleen in normal conditions and after acute blood loss. The largest amounts of detected secreted cytokines were the chemokines CCL2 and CCL3. We also detected some production of the cytokine IL-1b. The list of detected cytokines is in accord with the NanoString gene expression analysis–we detected *Ccl2*, *Ccl3* and *Il1b* gene expression and have not detected gene expression of the undetected cytokines by the Bio-Plex method (see Fig 4A). Moreover, the production of the chemokine CCL3 by Ter-119+ erythroid cells from the bone marrow was significantly lower after the acute blood loss compared to the normal bone marrow Ter-119+ erythroid cell CCL3 levels (Fig 6).

**Table 1. Differential gene expression analysis of the post-ABL erythroid cells.** We considered q-values < 0.01 and log2(FC) < -1 or log2(FC) > 1 significant.

| Genes | log2(FC) | q-value | log2(Mean) |
|---|---|---|---|
| Hlx | 1,87477 | 0,003304 | 7,00494703 |
| Ltb4r1 | 1,519352 | 0,000657 | 7,28639282 |
| Ctsg | 1,302794 | 0,003344 | 10,0653283 |
| Csf3r | 1,274738 | 0,001566 | 7,83427247 |
| Clec5a | 1,249117 | 0,003317 | 8,61593601 |
| Cdkn1a | 1,169783 | 0,001747 | 7,79369334 |
| Clec4e | 1,047907 | 0,016741 | 9,53992569 |
| Nfil3 | 0,896776 | 0,012434 | 7,37469849 |
| Nfkbia | 0,834517 | 0,000424 | 8,94117584 |
| Camp | 0,768596 | 0,004611 | 14,457602 |
| Cebpb | 0,760538 | 0,001692 | 9,69110485 |
| Vcam1 | 0,757209 | 0,039872 | 7,56620367 |
| Ifngr1 | 0,720312 | 0,003164 | 9,087193 |
| Itgb2 | 0,717999 | 0,00451 | 8,31901438 |
| S100a8 | 0,654607 | 0,016343 | 17,0360661 |
| Sell | 0,631787 | 0,021061 | 9,03527089 |
| Ptafr | 0,586265 | 0,001359 | 7,9906437 |
| Mapkapk2 | 0,563661 | 0,014714 | 8,34572552 |
| Cxcr4 | 0,552018 | 0,010638 | 10,4247994 |
| Ifnar2 | 0,490648 | 0,014312 | 8,25010048 |
| Plaur | 0,422957 | 0,05239 | 10,2709953 |
| Cd97 | 0,378523 | 0,002191 | 8,22042178 |
| Irf1 | 0,359004 | 0,05239 | 7,98611696 |
| Cfp | 0,322791 | 0,11648 | 8,35442409 |
| App | 0,310457 | 0,007493 | 8,31720621 |
| Itgam | 0,292408 | 0,049542 | 9,0953678 |
| Cybb | 0,2923 | 0,05239 | 11,2064301 |
| Litaf | 0,267387 | 0,031155 | 8,87761634 |
| Cd83 | 0,243612 | 0,049542 | 8,5578499 |
| C3 | 0,225728 | 0,086791 | 9,51335317 |
| Arhgdib | 0,216435 | 0,049542 | 12,0132289 |
| H2-Ea-ps | 0,196731 | 0,137257 | 9,43008862 |
| Cd44 | 0,194485 | 0,033312 | 9,67751557 |
| Fcgr2b | 0,188873 | 0,179545 | 9,18894842 |
| Stat3 | 0,179201 | 0,100932 | 8,52522078 |
| Ncf4 | 0,165842 | 0,06747 | 10,2195464 |
| Tyrobp | 0,149933 | 0,224617 | 11,7889501 |
| Ptprc | 0,10363 | 0,204569 | 9,37343616 |
| Fcgr3 | 0,077704 | 0,182963 | 7,99295713 |
| Xbp1 | 0,044349 | 0,302128 | 9,67757626 |
| Ebi3 | 0,043823 | 0,354737 | 8,04141273 |
| Ptpn6 | -0,007568 | 0,437307 | 8,92708144 |
| Mif | -0,019966 | 0,415805 | 8,41776078 |
| Il16 | -0,048724 | 0,37735 | 8,33101138 |
| Tgfbi | -0,070686 | 0,364214 | 8,4088589 |
| Trem1 | -0,070816 | 0,30342 | 9,27818248 |
| Fcer1g | -0,086386 | 0,274702 | 9,05310621 |

*(Continued)*

**Table 1.** (Continued)

| Genes | log2(FC) | q-value | log2(Mean) |
|---|---|---|---|
| Ifnar1 | -0,097318 | 0,203519 | 8,56088964 |
| S100a9 | -0,100654 | 0,32006 | 17,6473253 |
| Nfkbiz | -0,117015 | 0,324631 | 9,50509942 |
| Cd74 | -0,119386 | 0,281878 | 11,0092219 |
| Cd9 | -0,139032 | 0,193707 | 8,68109707 |
| Spn | -0,14196 | 0,100932 | 8,41126195 |
| H2-K1 | -0,215382 | 0,177752 | 9,03037983 |
| Jak2 | -0,229355 | 0,120807 | 8,56593721 |
| Il1b | -0,231095 | 0,071378 | 9,32521239 |
| Abcb10 | -0,237792 | 0,182963 | 9,33599829 |
| Prkcd | -0,26005 | 0,0814 | 8,61452084 |
| Ctss | -0,276268 | 0,198094 | 9,08573067 |
| Jak1 | -0,301748 | 0,000314 | 9,41180728 |
| Lilrb4 | -0,30381 | 0,140438 | 10,6373557 |
| Cd164 | -0,349859 | 0,001692 | 9,94146245 |
| Nfkb2 | -0,359722 | 0,013041 | 8,04800464 |
| Il17ra | -0,375208 | 0,05812 | 8,39730795 |
| Tgfb1 | -0,388059 | 0,010638 | 9,46170607 |
| Cd82 | -0,4315 | 0,049542 | 9,99165444 |
| Bcap31 | -0,435809 | 0,000657 | 8,68275083 |
| Ccl9 | -0,457302 | 0,054017 | 9,16465069 |
| Psmd7 | -0,471534 | 0,009418 | 8,88312017 |
| Stat6 | -0,490435 | 0,005473 | 7,95969364 |
| Syk | -0,513637 | 0,00598 | 9,49659107 |
| Ppbp | -0,513691 | 0,30342 | 6,66284178 |
| C1qa | -0,519964 | 0,148378 | 7,83628645 |
| Tnfrsf14 | -0,536492 | 0,087313 | 10,2727077 |
| Cd81 | -0,563596 | 0,003304 | 9,42096958 |
| Mapk1 | -0,572596 | 0,003317 | 9,51826507 |
| Fn1 | -0,576091 | 0,058418 | 8,34606556 |
| B2m | -0,578546 | 0,002599 | 12,4807427 |
| Casp3 | -0,581552 | 0,019845 | 8,91919995 |
| Tollip | -0,582838 | 0,002986 | 9,1485353 |
| Icam4 | -0,584225 | 0,052683 | 9,16782842 |
| Ccrl2 | -0,632056 | 0,001527 | 8,71454491 |
| Ube2l3 | -0,663464 | 0,003317 | 8,5241204 |
| Tfrc | -0,723355 | 0,044203 | 12,9232084 |
| Tal1 | -0,742908 | 0,046603 | 10,4227882 |
| Mapk14 | -0,759993 | 0,002191 | 8,52707519 |
| Psmb7 | -0,832746 | 0,000424 | 9,62649923 |
| Rela | -0,843845 | 0,001286 | 8,05856486 |
| Ets1 | -0,858636 | 0,003317 | 8,93834805 |
| Prim1 | -0,862025 | 0,004396 | 9,01273565 |
| Ccl2 | -0,912534 | 0,055319 | 8,04693254 |
| H2-Eb1 | -0,927035 | 0,007 | 7,9830276 |
| Ccl3 | -0,934065 | 0,012244 | 9,29790947 |
| Rae1 | -0,961163 | 0,002987 | 8,80530261 |

(*Continued*)

**Table 1.** (Continued)

| Genes | log2(FC) | *q*-value | log2(Mean) |
|---|---|---|---|
| *Chuk* | -0,967619 | 0,004611 | 7,63159427 |
| *Psmb5* | -0,973113 | 0,000314 | 8,64915252 |
| *Cd24a* | -1,024528 | 0,004396 | 14,1815281 |
| *H2-Aa* | -1,044709 | 0,003916 | 10,5056178 |
| *C1qbp* | -1,100559 | 0,003916 | 9,17332236 |
| *Psmc2* | -1,128281 | 0,000314 | 10,3143583 |
| *Irgm1* | -1,147005 | 0,000424 | 8,04646398 |
| *Tnfaip3* | -1,331172 | 0,000424 | 9,6579562 |
| *Cd36* | -1,363871 | 0,016741 | 9,89639128 |
| *Itgb1* | -1,380271 | 0,000482 | 9,68691756 |
| *Ms4a1* | -1,380661 | 0,002599 | 7,54883028 |
| *Ptpn2* | -1,394648 | 0,000432 | 7,71107503 |
| *Ctnnb1* | -1,44872 | 0,000424 | 9,49272111 |
| *Cd55* | -1,455002 | 0,003317 | 8,11573244 |
| *Pml* | -1,536634 | 0,003304 | 8,35742732 |
| *C1qb* | -1,55099 | 0,02576 | 8,1762979 |
| *Ccr2* | -1,564424 | 0,000424 | 8,25134501 |
| *Cd79b* | -1,581816 | 0,001692 | 8,14782409 |
| *Itga4* | -1,703999 | 0,002539 | 8,82683222 |
| *Il1r2* | -2,073272 | 0,000314 | 10,0327108 |
| *Cd2* | -2,363551 | 0,000424 | 8,05803935 |
| *Cxcl12* | -3,8068 | 0,000583 | 7,40617888 |
| *H2-Ab1* | -4,877545 | 0,000413 | 6,94301249 |

## Discussion

Our data allow us to say that acute blood loss has a powerful disturbing effect on hematopoiesis, especially erythropoiesis. With acute blood loss, blood leaks out of the body, therefore there is hypovolemia and loss of all blood cells without exception.

We noticed that during blood loss in mice, the content of erythroblasts decreases in the bone marrow and in the spleen, which is probably due to the acceleration of erythroblast maturation and the release of reticulocytes into the peripheral blood. Several arguments can be given in favor of this thesis by analyzing changes in the phenotypic forms of erythron in hematopoietic organs. As we noticed, the bone marrow and spleen react to acute blood loss in a similar way—by accelerating the maturation of the CD45– and CD45+ branches of Ter-119 +erythroid cells. It can be seen that in both the bone marrow and the spleen, during blood loss, the content of early maturation forms–proerythroblasts, basophilic erythroblasts—- significantly increases, and the content of late maturation forms—orthochromatophilic erythroblasts and reticulocytes drops drastically; apparently, reticulocytes do not remain in the hematopoietic organs for long, but are released into the peripheral blood as soon as possible.

Also, with blood loss in the bone marrow and spleen, the proportion of early forms—proerythroblasts, basophilic erythroblasts—increases, indicating accelerated proliferation and maturation of erythroid cells. However, the following difference can be noted: CD45+ and CD45- branches respond to acute blood loss differently, in particular different potency of CD45+ and CD45- polychromatic erythroblasts to division and maturation. We observed increased content of marrow CD45+ polychromatic erythroblasts after acute blood loss. Also, we noticed an increased content of splenic CD45+ polychromatic erythroblasts after acute blood loss.

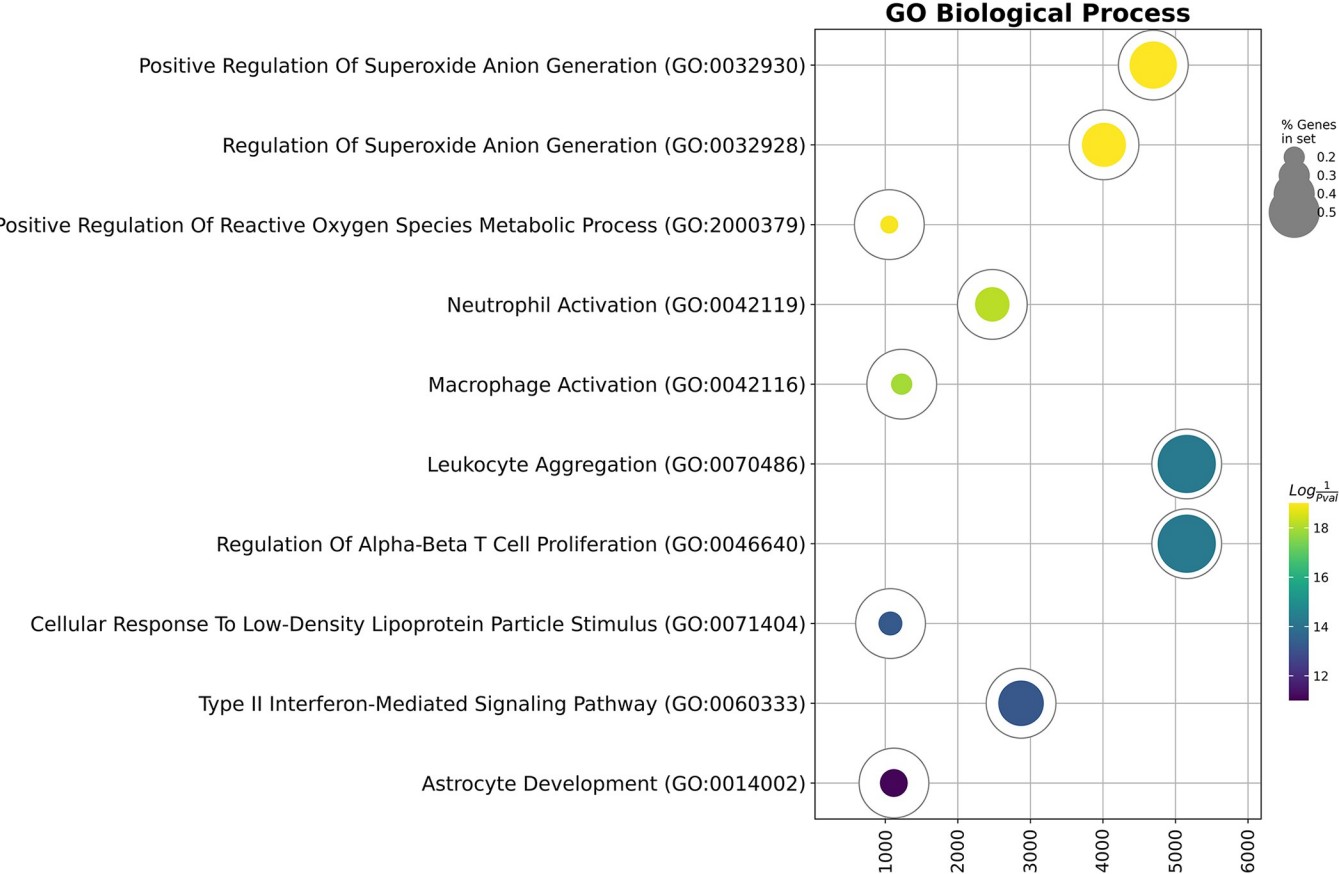

**Fig 5. Gene Ontology Biological Process overrepresentation analysis of the genes expressed in the post-ABL Ter-119⁺ erythroid cells, the yellow color corresponds to the lowest *q*-value, the deep purple color corresponds to the highest *q*-value, and the dot size reflects the percentage of genes included in the analysis relative to the full set of genes in the Gene Ontology Biological Process database.**

However, the content of CD45- polychromatic erythroblasts decreased in the marrow and in the spleen after acute blood loss. Hence, we have hypothesized that CD45+ polychromatic erythroblasts have the tendency to have more division and differentiation than CD45-

**Table 2. Gene Ontology Biological Process overrepresentation analysis of the genes expressed in the post-ABL Ter-119⁺ erythroid cells.**

| Term | Overlap | Q-value | Score | Genes |
|---|---|---|---|---|
| Positive Regulation Of Superoxide Anion Generation (GO:0032930) | 6/13 | 0.00000 | 4694 | *ITGAM, TGFB1, TYROBP, SYK, ITGB2, PRKCD* |
| Regulation Of Superoxide Anion Generation (GO:0032928) | 6/14 | 0.00000 | 4015 | *ITGAM, TGFB1, TYROBP, SYK, ITGB2, PRKCD* |
| Positive Regulation Of Reactive Oxygen Species Metabolic Process (GO:2000379) | 8/48 | 0.00000 | 1057 | *CDKN1A, ITGAM, TGFB1, TYROBP, SYK, ITGB2, PRKCD, MAPK14* |
| Neutrophil Activation (GO:0042119) | 6/18 | 0.00000 | 2477 | *FCER1G, TYROBP, SYK, PRKCD, CTSG, CAMP* |
| Macrophage Activation (GO:0042116) | 7/35 | 0.00000 | 1227 | *APP, C1QA, ITGAM, SYK, IFNGR1, ITGB2, JAK2* |
| Leukocyte Aggregation (GO:0070486) | 4/7 | 0.00000 | 5156 | *IL1B, S100A9, CD44, S100A8* |
| Regulation Of Alpha-Beta T Cell Proliferation (GO:0046640) | 4/7 | 0.00000 | 5156 | *SYK, IRF1, EBI3, TNFRSF14* |
| Cellular Response To Low-Density Lipoprotein Particle Stimulus (GO:0071404) | 5/22 | 0.00000 | 1073 | *FCER1G, SYK, CD81, ITGB2, CD9* |
| Type II Interferon-Mediated Signaling Pathway (GO:0060333) | 4/9 | 0.00000 | 2873 | *IFNGR1, IRF1, JAK2, JAK1* |
| Astrocyte Development (GO:0014002) | 4/15 | 0.00000 | 1120 | *APP, C1QA, IFNGR1, S100A9* |

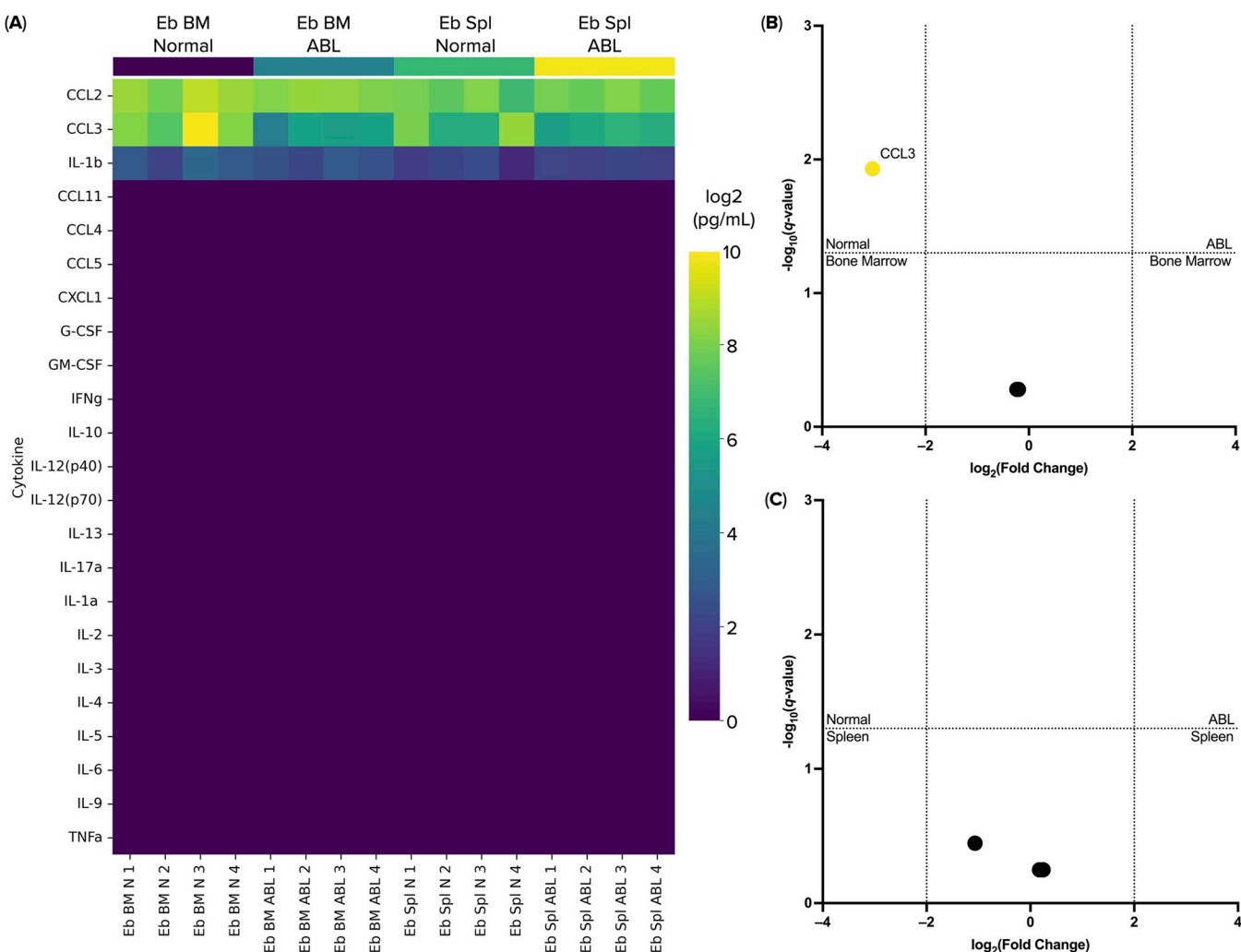

**Fig 6. Cytokine production profiles of the Ter-119+ erythroid cell.** (A) heat map of the cytokines secreted by Ter-119+ erythroid cells. Eb stands for Ter-119+ erythroid cells, Bm stands for Bone Marrow, Spl stands for Spleen, N stands for normal, ABL stands for post-acute blood loss; (B) Volcano-plot of the differentially secreted cytokines by Ter-119+ erythroid cells from the bone marrow; (C) Volcano-plot of the differentially secreted cytokines by Ter-119+ erythroid cells from the spleen.

polychromatic erythroblasts which behave like reticulocytes because CD45- polychromatic erythroblasts release to the bloodstream like reticulocytes and finish the maturation in the blood. We suppose the decreasing CD45- polychromatic erythroblasts content in the hemato-poietic organs connects with their exit to the bloodstream. But these suggestions need to be proofed. We can say that in conditions of blood loss when the body has a request for new blood cells, in particular, red blood cells, erythron responds with accelerated maturation of CD45– Ter-119+ erythroid cells to replenish lost red blood cells. Something similar was described by Shahbaz and co-authors [14]. With COVID-19, an increase in immature ery-throid cells occurs in the human bone marrow, and erythroblasts are also found in the periph-eral blood, and this phenomenon correlates with the severity of the infection. That is, under conditions of increasing tissue hypoxia with lung damage, when the body has a request for new red blood cells, increased proliferation of the erythroid sprout occurs, and even the release of nuclear erythroid cells into the bloodstream as a compensatory effect.

Gene expression analysis allowed us to delineate the list of cytokines expressed in Ter-119 + erythroid cells: *Ccl3*, *Tgfb1*, *Il1b*, *Cxcl12*, *Ccl2*, *Mif*, and *Il16*. These cytokines are represented by chemokines, proinflammatory IL1B, and tolerogenic TGFB1, which suggests different and multidirectional immunoregulatory functions for Ter-119+ erythroid cells—chemoattraction, immunosuppression, and initiation of inflammation (Fig 4B). We also previously demonstrated the expression of chemokine genes in human erythroid cells [15, 16]. Gene Ontology Biological Process analysis for genes active in Ter-119+ erythroid cells of the bone marrow after acute blood loss showed the presence of an active process of synthesis of reactive oxygen species in erythroid cells, which is consistent with literature data [17].

With blood loss in Ter-119+erythroid cells of the bone marrow, the expression of some genes increases, such as *Csf3r*, *Cdkn1a*, *Ctsg*, *Hlx*, *Clec5a*, and *Ltb4r1*. The *Clec5a* gene encodes the CLEC5A protein, which is a pathogen-associated molecular pattern (PAMP) recognition receptor [18], which may indicate the involvement of Ter-119+ erythroid cells in the process of innate immunity through sensing of PAMPs using CLEC5A and elimination of detected pathogens by the proteins Calprotectin (*S100a8* and *S100a9* genes) and Cathelicidin (*Camp* gene) [19] expressed by Ter-119+ erythroid cells (Fig 4B).

The *Csf3r* gene product is a G-CSF receptor expressed primarily by hematopoietic stem cells and myeloid lineage cells. This receptor is necessary for the rapid release of cells into the bloodstream during the mobilization of hematopoiesis [20]. Perhaps its increased expression in erythroblasts during blood loss is necessary to accelerate the maturation and further release of erythroid cells into the bloodstream.

The *Cdkn1a* (*p21*) gene encodes a protein inhibitor of cyclin-dependent kinase 1a, thereby preventing the cell from entering the G1 phase of the cell cycle, and maintaining the cell in the resting phase. During blood loss in bone marrow erythroblasts, we noticed a significant increase in the expression of the *Cdkn1a* gene, however, it must be said that the cell cycle of an erythroblast is very different from other cells, since erythropoiesis is a unique process in which each cell division is simultaneously associated with the stage of differentiation [21]. Elevated levels of *Cdkn1a* mediate erythroblast apoptosis in beta-thalassemia but do not improve erythropoiesis [22].

We also showed that during acute blood loss in erythroblasts, the expression of *Mif* does not change significantly. It is known that this cytokine interacts with its receptor CD74, the gamma chain of MHC class II, therefore their interaction is important for antigen presentation. It was shown that erythroblasts in all cases we studied express *Cd74*. Moreover, under any influences, as well as during ontogenesis, the expression of *Cd74*in erythroid cells does not change. It has also been shown that blockade or deletion of CD74 leads to the accumulation of hematopoietic stem cells in the bone marrow [23].

We also detected the expression of genes associated with the presentation of exogenous antigen in the MHC class II complex in the normal bone marrow murine Ter-119+ erythroid cells: *Ctsg*, *Cd74*, *H2-Aa*, and *H2-Ab1*, which suggests the possibility of presentation of exogenous antigen by mouse bone marrow erythroid cells. The presence of expression of exogenous antigen presentation genes in the MHC class II complex, coupled with the production of reactive oxygen species and their expression of antimicrobial genes *S100a8*, *S100a9*, and *Camp*, makes the transcriptome of Ter-119+ erythroid cells functionally similar to that of classical monocytes [24, 25]. The expression of the *H2-Aa* and *H2-Ab1* genes was down-regulated after acute blood loss, which suggests a disruption in the process of exogenous antigen presentation, provided that such a process is actually present in Ter-119+ erythroid cells.

The list of cytokines detected by the Bio-Plex method was in accord with the NanoString gene expression analysis–we have found both gene expression and cytokine secretion of CCL2, CCL3, and IL-1b. The production of chemokines and cytokines by Ter-119+ erythroid cells

undergo slight changes after the acute blood loss: bone marrow erythroblasts produce significantly less chemokine CCL3 compared to normal. The decreased CCL3 chemokine production may lead to a reduced chemoattraction of cells that express CCR1, CCR4, and CCR5 chemokine receptors. Some authors suggest the prognostic role of CCL3 in autoimmune diseases such as multiple sclerosis [26]. In light of erythropoiesis, elevated levels of CCL3 in the serum of patients with multiple myeloma have been described to suppress the maturation of erythroblasts into reticulocytes, leading to anemia [27]. It can be assumed that CCL3 has an indirect effect on erythropoiesis and, in our case, post-acute blood loss, its reduced production can be explained by the need for new red blood cells in the body and the acceleration of maturation of erythroid progenitor cells. Previously, in our secretomic analysis of the healthy adult human erythroid cell conditional media we have found the secretion of CCL22, CCL24, CXCL5, CXCL8, and MIF chemokines. This spectrum of chemokines could allow erythroid cells to attract neutrophils and eosinophils, which could help erythroid cells restrict the aforementioned cell types to the bone marrow in normal condition or initiate their migration to a site of extramedullary erythropoiesis. Maybe it makes chemokine secretion feature evolutionary conservative among erythroid cells [28].

## Conclusions

Acute blood loss stimulates terminal differentiation of erythroid cells in the murine bone marrow and spleen, with a decrease in the relative number of erythroid cells in the bone marrow, and also leads to the up-regulation of several immunity-related genes, including the pathogen sensor gene *Clec5a* in bone marrow erythroid cells, which could help bone marrow erythroid cells detect pathogens and eliminate them with the help of reactive oxygen species and antimicrobial proteins calprotectin and cathelicidin, the genes of which (*S100a8*, *S100a9*, and *Camp*) dominate in expression in bone marrow erythroid cells of mice.

## Supporting information

**S1 Graphical abstract.**
(TIF)

## Acknowledgments

Graphical Abstract was created using BioRender.

## Author Contributions

**Conceptualization:** Roman Perik-Zavodskii, Sergey Sennikov.

**Data curation:** Julia Shevchenko.

**Formal analysis:** Roman Perik-Zavodskii, Olga Perik-Zavodskaia, Saleh Alrhmoun, Marina Volynets.

**Investigation:** Kirill Nazarov.

**Methodology:** Julia Shevchenko.

**Project administration:** Sergey Sennikov.

**Software:** Roman Perik-Zavodskii, Olga Perik-Zavodskaia, Saleh Alrhmoun, Marina Volynets.

**Supervision:** Julia Shevchenko, Sergey Sennikov.

**Validation:** Kirill Nazarov.

**Visualization:** Roman Perik-Zavodskii, Olga Perik-Zavodskaia.

**Writing – original draft:** Kirill Nazarov, Roman Perik-Zavodskii.

**Writing – review & editing:** Kirill Nazarov, Roman Perik-Zavodskii.

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
