## [Decision Letter · Decision Letter 0]

26 Jun 2024

PONE-D-24-19541Acute blood loss in mice forces terminal differentiation of the CD45-negative erythroid cells and up-regulation of the pathogen-sensing gene Clec5a in bone marrow erythroid cellsPLOS ONE

Dear Dr. Sennikov,

Thank you for submitting your manuscript to PLOS ONE. After careful consideration, we feel that it has merit but does not fully meet PLOS ONE’s publication criteria as it currently stands. Therefore, we invite you to submit a revised version of the manuscript that addresses the points raised during the review process.

We look forward to receiving your revised manuscript.

Kind regards,

Gary S. Stein

Academic Editor

PLOS ONE

https://www.sciencedirect.com/science/article/pii/S0163725821001297?via%3Dihub

https://www.mdpi.com/1422-0067/24/9/8130

In your revision ensure you cite all your sources (including your own works), and quote or rephrase any duplicated text outside the methods section. Further consideration is dependent on these concerns being addressed.

Reviewers' comments:

Reviewer's Responses to Questions

**Comments to the Author**

1. Is the manuscript technically sound, and do the data support the conclusions?

Reviewer #1: Partly

2. Has the statistical analysis been performed appropriately and rigorously? 

Reviewer #1: Yes

3. Have the authors made all data underlying the findings in their manuscript fully available?

Reviewer #1: No

4. Is the manuscript presented in an intelligible fashion and written in standard English?

Reviewer #1: Yes

5. Review Comments to the Author

**Reviewer #1:** In this manuscript, Nazarov et al. aimed at investigating the immune function of the erythrocytes induced by acute blood loss (ABL). To this end, the authors collected the bone marrow and spleens of mice three days after blood drawn, and performed flow cytometry analysis of the different stages of erythrocytes, Nanostring gene expression profiling of immune response genes, and BioPlex analysis of multiple cytokines in the media conditioned with cultured erythrocytes. While the potential immune function of the erythrocytes upon ABL is interesting, this manuscript, at least in its current form, seems to be too preliminary and does not achieve its aim very well, because it lacks sufficient information for describing the background, rationale for experimental design, and inter-validation of the different parts of the results. Specific concerns include:

1. The notion that erythrocyte differentiation can be stimulated by ABL has been well documented (e.g., reviewed by Valent et al. Haematologica. 2018, 103(10):1593-1603; Srole and Ganz. J Cell Physiol. 2021, 236(7):4888-4901). Although the current study might provide more details, the authors should cite the previous studies and make necessary comparisons. Any new finding or discrepancy (e.g., the decrease of reticulocytes upon ABL, as shown in Fig. 3) should be described and explained in detail.

2. In Figs. 1 and 3, besides the statistical analysis results, representative results of flow cytometry analysis of the bone marrow and spleen samples upon ABL should also be provided for readers’ better understanding and for objective evaluation of the quality of the data.

3. In the “Acute blood loss leads to the up-regulation of the pathogen-sensing Clec5a gene and the down-regulation of the early erythroid cell genes Itga4 and Itgb1 in Ter-119+ erythroid cells” section, too many gene names were simply listed in the main text. It would be better to put them in a table, together with the values of their expression levels in different samples.

4. Clec5a seems to be just one of the many differentially expressed genes. Without more technical validation, determination of its protein level, or functional experiments to explain its role in the erythrocyte differentiation or immune response, it is not reasonable to emphasize Clec5a too much over other genes, let alone highlighting it in the title of the manuscript.

5. There are several problems regarding the BioPlex analysis of cytokine levels. First, as this assay was performed using the media of cultured erythrocytes, the results may not reflect the in vivo cytokine secretion functions of the ABL-stimulated cells. Second, as most of the cytokines seemed to be undetectable in this assay, it is questionable whether the viability and functions of the cells are comparable with those in vivo or fresh isolated. Third, there is no inter-validation between the gene expression profiling and the BioPlex analysis results. These problems should be explained or at least discussed by providing more supporting information.

6. PLOS authors have the option to publish the peer review history of their article (what does this mean?). If published, this will include your full peer review and any attached files.

Reviewer #1: **Yes: **Xiao-Jian Sun

---

## [Author Response · Author response to Decision Letter 0]

5 Aug 2024

Dear editor

Thank you for your notes and recommendations. 

1. We revised the manuscript and made it in accordance with PlosOne requirements.

2. Roman Perik-Zavodskii edited the language level in this manuscript. 

3. Indeed, there are some overlappings with our previous articles, but in this manuscript we added references in all places where we mentioned some from our previous articles 

4. All needed data are in supplementary files (row data in particular)

Dear reviewer

First of all. Thank you for your revision and notes about our manuscript. We wanna respond to your notes. 

1. We added some more information about erythropoiesis at acute blood loss conditions. Also, we wanna note that decreasing content of late erythron forms such as orthochomatic erythroblasts and reticulocytes connect with their exit into bloodstream as we suppose. Their exit can be consider like compensatory effect to replace loosed RBC after ABL.

2. Mentioned figures were replaced.

3. List of genes were replaced to the table.

4. We agree that Clec5a seems to be just one of the many differentially expressed genes. It need more pointed experiments and functional tests to proof role of this gene in erythron physiology. Now we biased the emphasis to chemokine CCL3, because its mRNA expression and protein production are detected in marrow erythroblasts. We suppose that erythroblasts are potent chemokine-producers and erythroblasts realize immunoregulatory effects via the chemokines. 

5. About validation Nanostring (mRNA expression) and Bioplex data (protein production). We agree that in vivo and in vitro results may be different, because in vitro is some idealization model, which facilitate our work and so on. But in vitro model allow see clear affects for example in erythroid cells. We wanna note that we used fresh-isolated erythroid cells for flow cytometry and for Nanostrung analysis. Viability of isolated cells assayed by trypane blue test. Cell cultures were used to next procedures if their viability were 95% or more. At Nanostring analysis we applied more strict criterions to diminish threshold signals and extract these genes which are expressed exactly. And we see that production of protein have accordance with mRNA expression (IL1b, CCL2, CCL3). Also CCL3 expression level is equal Tal1 or CD36 expression level approximately (Tal1 and CD36 are signature genes of erythroid lineage). Although we detected other cytokines and chemokines genes expression we couldn’t check their protein production level by erythroblasts because that molecules (MIF, TGFb1, CXCL12, IL16) didn’t presented in Bioplex panel.

---

## [Editor Report · Decision Letter 1]

12 Aug 2024

Acute blood loss in mice forces differentiation of both CD45-positive and CD45-negative erythroid cells and leads to a desreased CCL3 chemokine production by bone marrow erythroid cell

PONE-D-24-19541R1

Dear Dr. Sennikov,

We’re pleased to inform you that your manuscript has been judged scientifically suitable for publication and will be formally accepted for publication once it meets all outstanding technical requirements.

Kind regards,

Gary S. Stein

Academic Editor

PLOS ONE
---

## [Editor Report · Acceptance letter]

23 Aug 2024

PONE-D-24-19541R1 

PLOS ONE

Dear Dr. Sennikov, 

I'm pleased to inform you that your manuscript has been deemed suitable for publication in PLOS ONE. Congratulations! Your manuscript is now being handed over to our production team.

Kind regards, 

on behalf of

Dr. Gary S. Stein 

Academic Editor

PLOS ONE